# Parameter-Free Online Learning via Model Selection

**Dylan J. Foster**
Cornell University

**Satyen Kale**
Google Research

**Mehryar Mohri**
NYU and Google Research

**Karthik Sridharan**
Cornell University

## Abstract

We introduce an efficient algorithmic framework for model selection in online learning, also known as parameter-free online learning. Departing from previous work, which has focused on highly structured function classes such as nested balls in Hilbert space, we propose a generic meta-algorithm framework that achieves online model selection oracle inequalities under minimal structural assumptions. We give the first computationally efficient parameter-free algorithms that work in arbitrary Banach spaces under mild smoothness assumptions; previous results applied only to Hilbert spaces. We further derive new oracle inequalities for matrix classes, non-nested convex sets, and $\mathbb{R}^d$ with generic regularizers. Finally, we generalize these results by providing oracle inequalities for arbitrary non-linear classes in the online supervised learning model. These results are all derived through a unified meta-algorithm scheme using a novel "multi-scale" algorithm for prediction with expert advice based on random playout, which may be of independent interest.

## 1 Introduction

A key problem in the design of learning algorithms is the choice of the hypothesis set $\mathcal{F}$. This is known as the *model selection* problem. The choice of $\mathcal{F}$ is driven by inherent trade-offs. In the statistical learning setting, this can be analyzed in terms of the *estimation* and *approximation errors*. A richer or more complex $\mathcal{F}$ helps better approximate the Bayes predictor (smaller approximation error). On the other hand, a hypothesis set that is too complex may have too large a VC-dimension or have unfavorable Rademacher complexity, thereby resulting in looser guarantees on the difference between the loss of a hypothesis and that of the best-in class (large estimation error).

In the batch setting, this problem has been extensively studied with the main ideas originating in the seminal work of [40] and [39] and the principle of Structural Risk Minimization (SRM). This is typically formulated as follows: let $(\mathcal{F}_i)_{i \in \mathbb{N}}$ be an infinite sequence of hypothesis sets (or models); the problem consists of using the training sample to select a hypothesis set $\mathcal{F}_i$ with a favorable estimation-approximation trade-off and choosing the best hypothesis $f$ in $\mathcal{F}_i$.

If we had access to a hypothetical oracle informing us of the best choice of $i$ for a given instance, the problem would reduce to the standard one of learning with a fixed hypothesis set. Remarkably though, techniques such as SRM or similar penalty-based model selection methods return a hypothesis $f^*$ that enjoys finite-sample learning guarantees that are almost as favorable as those that would be obtained had an oracle informed us of the index $i^*$ of the best-in-class classifier's hypothesis set [39; 13; 36; 21; 4; 24]. Such guarantees are sometimes referred to as *oracle inequalities*. They can be derived even for data-dependent penalties [21; 4; 3].

Such results naturally raise the following questions in the online setting: can we develop an analogous theory of model selection in online learning? Can we design online algorithms for model selection with solutions benefiting from strong guarantees, analogous to the batch ones? Unlike the statistical setting, in online learning one cannot split samples to first learn the optimal predictor within each subclass and then later learn the optimal subclass choice.

A series of recent works on online learning provide some positive results along that direction. On the algorithmic side, [25; 27; 30; 31] present solutions that efficiently achieve model selection oracle inequalities for the important special case where $\mathcal{F}_1, \mathcal{F}_2, \ldots$ is a sequence of nested balls in a Hilbert space. On the theoretical side, a different line of work focusing on general hypothesis classes [14] uses martingale-based sequential complexity measures to show that, information-theoretically, one can obtain oracle inequalities in the online setting at a level of generality comparable to that of the batch statistical learning. However, this last result is not algorithmic.

The first approach that a familiar reader might think of for tackling the online model selection problem is to run for each $i$ an online learning algorithm that minimizes regret against $\mathcal{F}_i$, and then aggregate over these algorithms using the multiplicative weights algorithm for prediction with expert advice. This would work if all the losses or "experts" considered were uniformly bounded by a reasonably small quantity. However, in many reasonable problems — particularly those arising in the context of online convex optimization — the losses of predictors or experts for each $\mathcal{F}_i$ may grow with $i$. Using simple aggregation would scale our regret with the magnitude of the largest $\mathcal{F}_i$ and not the $i^*$ we want to compare against. This is the main technical challenge faced in this context, and one that we fully address in this paper.

Our results are based on a novel *multi-scale algorithm* for prediction with expert advice. This algorithm works in a situation where the different experts' losses lie in different ranges, and guarantees that the regret to each individual expert is adapted to the range of its losses. The algorithm can also take advantage of a given prior over the experts reflecting their importance. This general, abstract setting of prediction with expert advice yields online model selection algorithms for a host of applications detailed below in a straightforward manner.

First, we give efficient algorithms for model selection for nested linear classes that provide oracle inequalities in terms of the norm of the benchmark to which the algorithm's performance is compared. Our algorithm works for any norm, which considerably generalizes previous work [25; 27; 30; 31] and gives the first polynomial time online model selection for a number of online linear optimization settings. This includes online oracle inequalities for high-dimensional learning tasks such as online PCA and online matrix prediction. We then generalize these results even further by providing oracle inequalities for arbitrary non-linear classes in the online supervised learning model. This yields algorithms for applications such as online penalized risk minimization and multiple kernel learning.

## 1.1  Preliminaries

**Notation.**  For a given norm $\|\cdot\|$, let $\|\cdot\|_\star$ denote the dual norm. Likewise, for any function $F$, $F^\star$ will denote its Fenchel conjugate. For a Banach space $(\mathfrak{B}, \|\cdot\|)$, the dual is $(\mathfrak{B}^\star, \|\cdot\|_\star)$. We use $x_{1:n}$ as shorthand for a sequence of vectors $(x_1, \ldots, x_n)$. For such sequences, we will use $x_t[i]$ to denote the $t$th vector's $i$th coordinate. We let $e_i$ denote the $i$th standard basis vector. $\|\cdot\|_p$ denotes the $\ell_p$ norm, $\|\cdot\|_\sigma$ denotes the spectral norm, and $\|\cdot\|_\Sigma$ denotes the trace norm. For any $p \in [1, \infty]$, let $p'$ be such that $\frac{1}{p} + \frac{1}{p'} = 1$.

**Setup and goals.**  We work in two closely related settings: online convex optimization (Protocol 1) and online supervised learning (Protocol 2). In online convex optimization, the learner selects decisions from a convex subset $\mathcal{W}$ of some Banach space $\mathfrak{B}$. Regret to a comparator $w \in \mathcal{W}$ in this setting is defined as $\sum_{t=1}^n f_t(w_t) - \sum_{t=1}^n f_t(w)$.

Suppose $\mathcal{W}$ can be decomposed into sets $\mathcal{W}_1, \mathcal{W}_2, \ldots$. For a fixed set $\mathcal{W}_k$, the optimal regret, if one tailors the algorithm to compete with $\mathcal{W}_k$, is typically characterized by some measure of intrinsic complexity of the class (such as Littlestone's dimension [5] and sequential Rademacher complexity [33]), denoted $\mathbf{Comp}_n(\mathcal{W}_k)$. We would like to develop algorithms that predict a sequence $(w_t)$ such that

$$\sum_{t=1}^n f_t(w_t) - \min_{w \in \mathcal{W}_k} \sum_{t=1}^n f_t(w) \le \mathbf{Comp}_n(\mathcal{W}_k) + \mathbf{Pen}_n(k) \quad \forall k. \tag{1}$$

This equation is called an *oracle inequality* and states that the performance of the sequence $(w_t)$ matches that of a comparator that minimizes the bias-variance tradeoff $\min_k \{\min_{w \in \mathcal{W}_k} \sum_{t=1}^n f_t(w) + \mathbf{Comp}_n(\mathcal{W}_k)\}$, up to a penalty $\mathbf{Pen}_n(k)$ whose scale ideally matches that of $\mathbf{Comp}_n(\mathcal{W}_k)$. We shall see shortly that ensuring that the scale of $\mathbf{Pen}_n(k)$ does

---

**Protocol 1** Online Convex Optimization

**for** $t = 1, \ldots, n$ **do**
  Learner selects strategy $q_t \in \Delta(\mathcal{W})$ for convex decision set $\mathcal{W}$.
  Nature selects convex loss $f_t \colon \mathcal{W} \to \mathbb{R}$.
  Learner draws $w_t \sim q_t$ and incurs loss $f_t(w_t)$.
**end for**

---

indeed match is the core technical challenge in developing online oracle inequalities for commonly used classes.

In the supervised learning setting we measure regret against a benchmark class $\mathcal{F} = \bigcup_{k=1}^{\infty} \mathcal{F}_k$ of functions $f \colon \mathcal{X} \to \mathbb{R}$, where $\mathcal{X}$ is some abstract context space, also called feature space. In this case, the desired oracle inequality has the form:

$$\sum_{t=1}^{n} \ell(\hat{y}_t, y_t) - \inf_{f \in \mathcal{F}_k} \sum_{t=1}^{n} \ell(f(x_t), y_t) \leq \mathbf{Comp}_n(\mathcal{F}_k) + \mathbf{Pen}_n(k) \quad \forall k. \tag{2}$$

---

**Protocol 2** Online Supervised Learning

**for** $t = 1, \ldots, n$ **do**
  Nature provides $x_t \in \mathcal{X}$.
  Learner selects randomized strategy $q_t \in \Delta(\mathbb{R})$.
  Nature provides outcome $y_t \in \mathcal{Y}$.
  Learner draws $\hat{y}_t \sim q_t$ and incurs loss $\ell(\hat{y}_t, y_t)$.
**end for**

---

## 2 Online Model Selection

### 2.1 The need for multi-scale aggregation

Let us briefly motivate the main technical challenge overcome by the model selection approach we consider. The most widely studied oracle inequality in online learning has the following form

$$\sum_{t=1}^{n} f_t(w_t) - \sum_{t=1}^{n} f_t(w) \leq O\Big((\|w\|_2 + 1)\sqrt{n \cdot \log((\|w\|_2 + 1)n)}\Big) \quad \forall w \in \mathbb{R}^d. \tag{3}$$

In light of (1), a *model selection* approach to obtaining this inequality would be to split the set $\mathcal{W} = \mathbb{R}^d$ into $\ell_2$ norm balls of doubling radius, i.e. $\mathcal{W}_k = \{w \mid \|w\|_2 \leq 2^k\}$. A standard fact [15] is that such a set has $\mathbf{Comp}_n(\mathcal{W}_k) = 2^k \sqrt{n}$ if one optimizes over it using Mirror Descent, and so obtaining the oracle inequality (1) is sufficient to recover (3), so long as $\mathbf{Pen}_n(k)$ is not too large relative to $\mathbf{Comp}_n(\mathcal{W}_k)$.

Online model selection is fundamentally a problem of prediction with expert advice [8], where the experts correspond to the different model classes one is choosing from. Our basic meta-algorithm, MULTISCALEFTPL (Algorithm 3), operates in the following setup. The algorithm has access to a finite number, $N$, of experts. In each round, the algorithm is required to choose one of the $N$ experts. Then the losses of all experts are revealed, and the algorithm incurs the loss of the chosen expert.

The twist from the standard setup is that the losses of all the experts are *not* uniformly bounded in the same range. Indeed, for the setup described for the oracle inequality (3), class $\mathcal{W}_k$ will produce predictions with norm as large as $2^k$. Therefore, here, we assume that expert $i$ incurs losses in the range $[-c_i, c_i]$, for some known parameter $c_i \geq 0$. The goal is to design an online learning algorithm whose regret to expert $i$ scales with $c_i$, rather than $\max_i c_i$, which is what previous algorithms for learning from expert advice (such as the standard multiplicative weights strategy or AdaHedge [12]) would achieve. Indeed, any regret bound scaling in $\max_i c_i$ will be far too large to achieve (3), as the term $\mathbf{Pen}_n(k)$ will dominate. This new type of scale-sensitive regret bound, achieved by our algorithm MULTISCALEFTPL, is stated below.

**Algorithm 3**

---

    **procedure** MULTISCALEFTPL$(c, \pi)$         $\triangleright$ Scale vector $c$ with $c_i \geq 1$, prior distribution $\pi$.

        **for** time $t = 1, \ldots, n$: **do**

                Draw sign vectors $\sigma_{t+1}, \ldots, \sigma_n \in \{\pm 1\}^N$ each uniformly at random.

                Compute distribution

$$p_t(\sigma_{t+1:n}) = \underset{p \in \Delta_N}{\arg\min} \sup_{g_t : |g_t[i]| \leq c_i} \left[ \langle p, g_t \rangle + \sup_{i \in [N]} \left[ -\sum_{s=1}^{t} \langle e_i, g_s \rangle + 4 \sum_{s=t+1}^{n} \sigma_s[i] c_i - B(i) \right] \right],$$

                where $B(i) = 5c_i \sqrt{n \log\left(4c_i^2 n / \pi_i\right)}$.

                Play $i_t \sim p_t$.

                Observe loss vector $g_t$.

        **end for**

    **end procedure**

---

**Theorem 1.** *Suppose the loss sequence $(g_t)_{t \leq n}$ satisfies $|g_t[i]| \leq c_i$ for a sequence $(c_i)_{i \in [N]}$ with each $c_i \geq 1$. Let $\pi \in \Delta_N$ be a given prior distribution on the experts. Then, playing the strategy $(p_t)_{t \leq n}$ given by Algorithm 3, MULTISCALEFTPL yields the following regret bound:*[1]

$$\mathbb{E}\left[ \sum_{t=1}^{n} \langle e_{i_t}, g_t \rangle - \sum_{t=1}^{n} \langle e_i, g_t \rangle \right] \leq O\left( c_i \sqrt{n \log(nc_i/\pi_i)} \right) \quad \forall i \in [N]. \tag{4}$$

The proof of the theorem is deferred to Appendix A in the supplementary material due to space constraints. Briefly, the proof follows the technique of adaptive relaxations from [14]. It relies on showing that the following function of the first $t$ loss vectors $g_{1:t}$ is an admissible relaxation (see [14] for definitions):

$$\mathbf{Rel}(g_{1:t}) \triangleq \mathbb{E}_{\sigma_{t+1}, \ldots, \sigma_n \in \{\pm 1\}^N} \sup_i \left[ -\sum_{s=1}^{t} \langle e_i, g_s \rangle + 4 \sum_{s=t+1}^{n} \sigma_s[i] c_i - B(i) \right].$$

This implies that if we play the strategy $(p_t)_{t \leq n}$ given by Algorithm 3, the regret to the $i$th expert is bounded by $B(i) + \mathbf{Rel}(\cdot)$, where $\mathbf{Rel}(\cdot)$ indicates the $\mathbf{Rel}$ function applied to an empty sequence of loss vectors. As a final step, we bound $\mathbf{Rel}(\cdot)$ as $O(1)$ using a probabilistic maximal inequality (Lemma 2 in the supplementary material), yielding the bound (4). Compared to related FTPL algorithms [34], the analysis is surprisingly delicate, as additive $c_i$ factors can spoil the desired regret bound (4) if the $c_i$s differ by orders of magnitude.

The min-max optimization problem in MULTISCALEFTPL can be solved in polynomial-time using linear programming — see Appendix A.1 in the supplementary material for a full discussion.

In related work, [7] simultaneously developed a multi-scale experts algorithm which could also be used in our framework. Their regret bound has sub-optimal dependence on the prior distribution over experts, but their algorithm is more efficient and is able to obtain multiplicative regret guarantees.

## 2.2 Online convex optimization

One can readily apply MULTISCALEFTPL for online optimization problems whenever it is possible to bound the losses of the different experts a-priori. One such application is to online convex optimization, where each "expert" is a a particular OCO algorithm, and for which such a bound can be obtained via appropriate bounds on the relevant norms of the parameter vectors and the gradients of the loss functions. We detail this application — which yields algorithms for parameter-free online learning and more — below. All of the algorithms in this section are derived using a unified meta-algorithm strategy MULTISCALEOCO.

The setup is as follows. We have access to $N$ sub-algorithms, denoted $\text{ALG}_i$ for $i \in [N]$. In round $t$, each sub-algorithm $\text{ALG}_i$ produces a prediction $w_t^i \in \mathcal{W}_i$, where $\mathcal{W}_i$ is a set in a vector space $V$ over $\mathbb{R}$ containing 0. Our meta-algorithm is then required to choose one of the predictions $w_t^i$. Then, a loss function $f_t : V \to \mathbb{R}$ is revealed, whereupon $\text{ALG}_i$ incurs loss $f_t(w_t^i)$, and the meta-algorithm suffers the loss of the chosen prediction. We make the following assumption on the sub-algorithms:

**Assumption 1.** The sub-algorithms satisfy the following conditions:

- For each $i \in [N]$, there is an associated norm $\|\cdot\|_{(i)}$ such that $\sup_{w \in \mathcal{W}_i} \|w\|_{(i)} \le R_i$.

- For each $i \in [N]$, the sequence of functions $f_t$ are $L_i$-Lipschitz on $\mathcal{W}_i$ with respect to $\|\cdot\|_{(i)}$.

- For each sub-algorithm $\text{ALG}_i$, the iterates $(w_t^i)_{t \le n}$ enjoy a regret bound $\sum_{t=1}^n f_t(w_t^i) - \inf_{w \in \mathcal{W}_i} \sum_{t=1}^n f_t(w) \le \mathbf{Reg}_n(i)$, where $\mathbf{Reg}_n(i)$ may be data- or algorithm-dependent.

---

**Algorithm 4**

---

**procedure** MULTISCALEOCO$(\{\text{ALG}_i, R_i, L_i\}_{i \in [N]}, \pi)$       $\triangleright$ Collection of sub-algorithms, prior $\pi$.
    $c \leftarrow (R_i \cdot L_i)_{i \in [N]}$                                                $\triangleright$ Sub-algorithm scale parameters.
    **for** $t = 1, \ldots, n$ **do**
        $w_t^i \leftarrow \text{ALG}_i(\tilde{f}_1, \ldots, \tilde{f}_{t-1})$ for each $i \in \mathcal{A}$.
        $i_t \leftarrow \text{MULTISCALEFTPL}[c, \pi](g_1, \ldots, g_{t-1})$.
        Play $w_t = w_t^{i_t}$.
        Observe loss function $f_t$ and let $\tilde{f}_t(w) = f_t(w) - f_t(0)$.
        $g_t \leftarrow \big(\tilde{f}_t(w_t^i)\big)_{i \in [N]}$.
    **end for**
**end procedure**

---

In most applications, $\mathcal{W}_i$ will be a convex set and $f_t$ a convex function; this convexity is not necessary to prove a regret bound for the meta-algorithm. We simply need boundedness of the set $\mathcal{W}_i$ and Lipschitzness of the functions $f_t$, as specified in Assumption 1. This assumption implies that for any $i$, we have $|f_t(w) - f_t(0)| \le R_i L_i$ for any $w \in \mathcal{W}_i$. Thus, we can design a meta-algorithm for this setup by using MULTISCALEFTPL with $c_i = R_i L_i$, which is precisely what is described in Algorithm 4. The following theorem provides a bound on the regret of MULTISCALEOCO; a direct consequence of Theorem 1.

**Theorem 2.** *Without loss of generality, assume that $R_i L_i \ge 1^2$. Suppose that the inputs to Algorithm 4 satisfy Assumption 1. Then the iterates $(w_t)_{t \le n}$ returned by Algorithm 4 follow the regret bound*

$$\mathbb{E}\left[\sum_{t=1}^n f_t(w_t) - \inf_{w \in \mathcal{W}_i} \sum_{t=1}^n f_t(w)\right] \le \mathbb{E}[\mathbf{Reg}_n(i)] + O\left(R_i L_i \sqrt{n \log(R_i L_i n / \pi_i)}\right) \quad \forall i \in [N]. \quad (5)$$

Theorem 2 shows that if we use Algorithm 4 to aggregate the iterates produced by a collection of sub-algorithms $(\text{ALG}_i)_{i \in [N]}$, the regret against any sub-algorithm $i$ will only depend on that algorithm's scale, not the regret of the worst sub-algorithm.

**Application 1: Parameter-free online learning in uniformly convex Banach spaces.** As the first application of our framework, we give a generalization of the parameter-free online learning bounds found in [25; 27; 30; 31; 10] from Hilbert spaces to arbitrary uniformly convex Banach spaces. Recall that a Banach space $(\mathfrak{B}, \|\cdot\|)$ is $(2, \lambda)$-uniformly convex if $\frac{1}{2}\|\cdot\|^2$ is $\lambda$-strongly convex with respect to itself [32]. Our algorithm obtains a generalization of the oracle inequality (3) for any uniformly convex $(\mathfrak{B}, \|\cdot\|)$ by running multiple instances of *Mirror Descent* — the workhorse of online convex optimization — and aggregating their iterates using MULTISCALEOCO. This strategy is thus efficient whenever Mirror Descent can be implemented efficiently. The collection of sub-algorithms used by MULTISCALEOCO, which was alluded to at the beginning of this section is as follows: For each $1 \le i \le N := n + 1$, set $R_i = e^{i-1}$, $L_i = L$, $\mathcal{W}_i = \{w \in \mathfrak{B} \mid \|w\| \le R_i\}$, $\eta_i = \frac{R_i}{L}\sqrt{\frac{\lambda}{n}}$, and $\text{ALG}_i = \text{MIRRORDESCENT}(\eta_i, \mathcal{W}_i, \|\cdot\|^2)$. Finally, set $\pi = \text{Uniform}([n+1])$.

Mirror Descent is reviewed in detail in Appendix A.2 in the supplementary material, but the only feature of its performance of importance to our analysis is that, when configured as described above, the

iterates $(w_t^i)_{t \le n}$ produced by $\text{ALG}_i$ specified above will satisfy $\sum_{t=1}^n f_t(w_t^i) - \inf_{w \in \mathcal{W}_i} \sum_{t=1}^n f_t(w) \le O(R_i L \sqrt{\lambda n})$ on any sequence of losses that are $L$-Lipschitz with respect to $\|\cdot\|_\star$. Using just this simple fact, combined with the regret bound for MULTISCALEOCO and a few technical details in Appendix A.2, we can deduce the following parameter-free learning oracle inequality:

**Theorem 3** (Oracle inequality for uniformly convex Banach spaces)**.** *The iterates* $(w_t)_{t \le n}$ *produced by* MULTISCALEOCO *on any $L$-Lipschitz (w.r.t. $\|\cdot\|_\star$) sequence of losses* $(f_t)_{t \le n}$ *satisfy*

$$\mathbb{E}\left[\sum_{t=1}^n f_t(w_t) - \sum_{t=1}^n f_t(w)\right] \le O\left(L \cdot (\|w\| + 1)\sqrt{n \cdot \log((\|w\| + 1)Ln)/\lambda}\right) \quad \forall w \in \mathfrak{B}. \quad (6)$$

Note that the above oracle inequality applies for **any uniformly convex norm** $\|\cdot\|$. Previous results only obtain bounds of this form efficiently when $\|\cdot\|$ is a Hilbert space norm. As is standard for such oracle inequality results, the bound is weaker than the optimal bound if $\|w\|$ were selected in advance, but only by a mild $\sqrt{\log((\|w\| + 1)Ln)}$ factor.

**Proposition 1.** The algorithm can be implemented in time $O(T_{\text{MD}} \cdot \text{poly}(n))$ per iteration, where $T_{\text{MD}}$ is the time complexity of a single Mirror Descent update.

In the example above, the $(2, \lambda)$-uniform convexity condition was mainly chosen for familiarity. The result can easily be generalized to related notions such as $q$-uniform convexity (see [37]). More generally, the approach can be used to derive oracle inequalities with respect to general strongly convex regularizer $\mathcal{R}$ defined over the space $\mathcal{W}$. Such a bound would have the form $O\left(L \cdot \sqrt{n(\mathcal{R}(w) + 1) \cdot \log((\mathcal{R}(w) + 1)n)}\right)$ for typical choices of $\mathcal{R}$.

This example captures well-known *quantile bounds* [22] when one takes $\mathcal{R}$ to be the KL-divergence and $\mathcal{W}$ to be the simplex, or, in the matrix case, takes $\mathcal{R}$ to be the quantum relative entropy and $\mathcal{W}$ to be the set of density matrices, as in [18].

**Application 2: Oracle inequality for many $\ell_p$ norms.** It is instructive to think of MULTISCALE-OCO as executing a (scale-sensitive) online analogue of the structural risk minimization principle. We simply specify a set of subclasses and a prior $\pi$ specifying the importance of each subclass, and we are guaranteed that the algorithm's performance matches that of each sub-class, plus a penalty depending on the prior weight placed on that subclass. The advantage of this approach is that the nested structure used in the Theorem 3 is completely inessential. This leads to the exciting prospect of developing parameter-free algorithms over new and exotic set systems. One such example is given now: The MULTISCALEOCO framework allows us to obtain an oracle inequality with respect to *many $\ell_p$ norms in $\mathbb{R}^d$ simultaneously*. To the best of our knowledge all previous works on parameter-free online learning have only provided oracle inequalities for a single norm.

**Theorem 4.** *Fix $\delta > 0$. Suppose that the loss functions* $(f_t)_{t \le n}$ *are $L_p$-Lipschitz w.r.t. $\|\cdot\|_{p'}$ for each $p \in [1 + \delta, 2]$. Then there is a computationally efficient algorithm that guarantees regret*

$$\mathbb{E}\left[\sum_{t=1}^n f_t(w_t) - \sum_{t=1}^n f_t(w)\right] \le O\left((\|w\|_p + 1)L_p\sqrt{n \log((\|w\|_p + 1)L_p \log(d)n)/(p-1)}\right) \quad (7)$$

*for all $w \in \mathbb{R}^d$ and all $p \in [1 + \delta, 2]$.*

The configuration in the above theorem is described in full in Appendix A.2 in the supplementary material. This strategy can be trivially extended to handle $p$ in the range $(2, \infty)$. The inequality holds for $p \ge 1 + \delta$ rather than for $p \ge 1$ because the $\ell_1$ norm is not uniformly convex, but this is easily rectified by changing the regularizer at $p = 1$; we omit this for simplicity of presentation.

We emphasize that the choice of $\ell_p$ norms for the result above was somewhat arbitrary — any finite collection of norms will also work. For example, the strategy can also be applied to matrix optimization over $\mathbb{R}^{d \times d}$ by replacing the $\ell_p$ norm with the Schatten $S_p$ norm. The Schatten $S_p$ norm has strong convexity parameter on the order of $p - 1$ (which matches the $\ell_p$ norm up to absolute constants [2]) so the only change to practical change to the setup in Theorem 4 will be the running time $T_{\text{MD}}$. Likewise, the approach applies to $(p, q)$-group norms as used in multi-task learning [20].

**Application 3: Adapting to rank for online PCA** For the online PCA task, the learner predicts from a class $\mathcal{W}_k = \left\{W \in \mathbb{R}^{d \times d} \mid W \ge 0, \|W\|_\sigma \le 1, \langle W, I \rangle = k\right\}$. For a fixed value of $k$, such a class is

a convex relaxation of the set of all rank $k$ projection matrices. After producing a prediction $W_t$, we experience affine loss functions $f_t(W_t) = \langle I - W_t, Y_t \rangle$, where $Y_t \in \mathcal{Y} := \{Y \in \mathbb{R}^{d \times d} \mid Y \geq 0, \|Y\|_\sigma \leq 1\}$. We leverage an analysis of online PCA due to [29] together with MULTISCALEOCO to derive an algorithm that competes with many values of the rank simultaneously. This gives the following result:

**Theorem 5.** *There is an efficient algorithm for Online PCA with regret bound*

$$\mathbb{E}\left[\sum_{t=1}^n \langle I - W_t, Y_t \rangle - \min_{\substack{W \text{ projection} \\ \text{rank}(W)=k}} \sum_{t=1}^n \langle I - W, Y_t \rangle\right] \leq \widetilde{O}\big(k\sqrt{n}\big) \quad \forall k \in [d/2].$$

For a fixed value of $k$, the above bound is already optimal up to log factors, but it holds for all $k$ simultaneously.

**Application 4: Adapting to norm for Matrix Multiplicative Weights** In the MATRIX MULTIPLICATIVE WEIGHTS setting [1] we consider hypothesis classes of the form $\mathcal{W}_r = \{W \in \mathbb{R}^{d \times d} \mid W \geq 0, \|W\|_\Sigma \leq r\}$. Losses are given by $f_t(W) = \langle W, Y_t \rangle$, where $\|Y_t\|_\sigma \leq 1$. For a fixed value of $r$, the well-known MATRIX MULTIPLICATIVE WEIGHTS strategy has regret against $\mathcal{W}_r$ bounded by $O(r\sqrt{n \log d})$. Using this strategy for fixed $r$ as a sub-algorithm for MULTISCALE-OCO, we achieve the following oracle inequality efficiently:

**Theorem 6.** *There is an efficient matrix prediction strategy with regret bound*

$$\mathbb{E}\left[\sum_{t=1}^n \langle W_t, Y_t \rangle - \sum_{t=1}^n \langle W, Y_t \rangle\right] \leq (\|W\|_\Sigma + 1)\sqrt{n \log d \log((\|W\|_\Sigma + 1)n))} \quad \forall W \geq 0. \qquad (8)$$

**A remark on efficiency** All of our algorithms that provide bounds of the form (6) instantiate $O(n)$ experts with MULTISCALEFTPL because, in general, the worst case $w$ for achieving (6) can have norm as large as $e^n$. If one has an a priori bound — say $B$ — on the range at which each $f_t$ attains its minimum, then the number of experts be reduced to $O(\log(B))$.

## 2.3 Supervised learning

We now consider the online supervised learning setting (Protocol 2), with the goal being to compete with a sequence of hypothesis classes $(\mathcal{F}_k)_{k \in [N]}$ simultaneously. Working in this setting makes clear a key feature of the meta-algorithm approach we have adopted: **We can efficiently obtain online oracle inequalities for arbitrary nonlinear function classes** — so long as we have an efficient algorithm for each $\mathcal{F}_k$.

We obtain a supervised learning meta-algorithm by simply feeding the observed losses $\ell(\cdot, y_t)$ (which may even be non-convex) to the meta-algorithm MULTISCALEFTPL in the same fashion as MULTISCALEOCO. The resulting strategy, which is described in detail in Appendix A.3 for completeness, is called MULTISCALELEARNING. We make the following assumptions analogous to Assumption 1, which lead to the performance guarantee for MULTISCALELEARNING given in Theorem 7 below.

**Assumption 2.** The sub-algorithms used by MULTISCALELEARNING satisfy the following conditions:

- For each $i \in [N]$, the iterates $(\hat{y}_t^i)_{t \leq n}$ produced by sub-algorithm ALG$_i$ satisfy $|\hat{y}_t^i| \leq R_i$.

- For each $i \in [N]$, the function $\ell(\cdot, y_t)$ is $L_i$-Lipschitz on $[-R_i, R_i]$.

- For each sub-algorithm ALG$_i$, the iterates $(\hat{y}_t^i)_{t \leq n}$ enjoy a regret bound $\sum_{t=1}^n \ell(\hat{y}_t^i, y_t) - \inf_{f \in \mathcal{F}_i} \sum_{t=1}^n \ell(f(x_t), y_t) \leq \mathbf{Reg}_n(i)$, where $\mathbf{Reg}_n(i)$ may be data- or algorithm-dependent.

**Theorem 7.** *Suppose that the inputs to Algorithm 5 satisfy Assumption 2. Then the iterates $(\hat{y}_t)_{t \leq n}$ produced by the algorithm enjoy the regret bound*

$$\mathbb{E}\left[\sum_{t=1}^n \ell(\hat{y}_t^i, y_t) - \inf_{f \in \mathcal{F}_i} \sum_{t=1}^n \ell(f(x_t), y_t)\right] \leq \mathbb{E}[\mathbf{Reg}_n(i)] + O\big(R_i L_i \sqrt{n \log(R_i L_i n / \pi_i)}\big) \quad \forall i \in [N]. \qquad (9)$$

**Online penalized risk minimization** In the statistical learning setting, oracle inequalities for arbitrary sequences of hypothesis classes $\mathcal{F}_1, \ldots, \mathcal{F}_N$ are readily available. Such inequalities are typically stated in terms of complexity parameters for the classes $(\mathcal{F}_k)$ such as VC dimension or Rademacher complexity. For the online learning setting, it is well-known that *sequential Rademacher complexity* $\mathbf{Rad}_n(\mathcal{F})$ provides a sequential counterpart to these complexity measures [33], meaning that it generically characterizes the minimax optimal regret for Lipschitz losses. We will obtain an oracle inequality in terms of this parameter.

**Assumption 3.** The sequence of hypothesis classes $\mathcal{F}_1, \ldots, \mathcal{F}_N$ are such that

1. There is an efficient algorithm $\mathrm{ALG}_k$ producing iterates $(\hat{y}_t^k)_{t \leq n}$ satisfying $\sum_{t=1}^n \ell(\hat{y}_t^k, y_t) - \inf_{f \in \mathcal{F}_k} \sum_{t=1}^n \ell(f(x_t), y_t) \leq C \cdot L \cdot \mathbf{Rad}_n(\mathcal{F}_k)$ for any $L$-Lipschitz loss, where $C$ is some constant. (an algorithm with this regret is always guaranteed to exist, but may not be efficient).

2. Each $\mathcal{F}_k$ has output range $[-R_k, R_k]$, where $R_k \geq 1$ without loss of generality.

3. $\mathbf{Rad}_n(\mathcal{F}_k) = \Omega(R_k \sqrt{n})$ — this is obtained by most non-trivial classes.

**Theorem 8** (Online penalized risk minimization). *Under Assumption 3 there is an efficient (in $N$) algorithm that achieves the following regret bound for any $L$-Lipschitz loss:*

$$\mathbb{E}\left[\sum_{t=1}^n \ell(\hat{y}_t, y_t) - \inf_{f \in \mathcal{F}_k} \sum_{t=1}^n \ell(f(x_t), y_t)\right] \leq O\left(L \cdot \mathbf{Rad}_n(\mathcal{F}_k) \cdot \sqrt{\log(L \cdot \mathbf{Rad}_n(\mathcal{F}_k) \cdot k)}\right) \quad \forall k \in [N]. \quad (10)$$

As in the previous section, one can derive tighter regret bounds and more efficient (e.g. sublinear in $N$) algorithms if $\mathcal{F}_1, \mathcal{F}_2, \ldots$ are nested.

### Application: Multiple kernel learning

**Theorem 9.** *Let $\mathcal{H}_1, \ldots, \mathcal{H}_N$ be reproducing kernel Hilbert spaces for which each $\mathcal{H}_k$ has a kernel $\mathbf{K}$ such that $\sup_{x \in \mathcal{X}} \sqrt{\mathbf{K}(x, x)} \leq B_k$. Then there is an efficient learning algorithm that guarantees*

$$\mathbb{E}\left[\sum_{t=1}^n \ell(\hat{y}_t, y_t) - \sum_{t=1}^n \ell(f(x_t), y_t)\right] \leq O\left(LB_k(\|f\|_{\mathcal{H}_k} + 1)\sqrt{\log(LB_k kn(\|f\|_{\mathcal{H}_k} + 1))}\right) \quad \forall k, \forall f \in \mathcal{H}_k$$

*for any $L$-Lipschitz loss, whenever an efficient algorithm is available for the norm ball in each $\mathcal{H}_k$.*

## 3 Discussion and Further Directions

**Related work** There are two directions in parameter-free online learning that have been explored extensively. The first considers bounds of the form (3); namely, the Hilbert space version of the more general setting explored in Section 2.2. Beginning with [26], which obtained a slightly looser rate than (3), research has focused on obtaining tighter dependence on $\|w\|_2$ and $\log(n)$ in this type of bound [25; 27; 30; 31]; all of these algorithms run in linear time per update step. Recent work [10; 11] has extended these results to the case where the Lipschitz constant is not known in advance. These works give lower bounds for general norms, but only give efficient algorithms for Hilbert spaces. Extending Algorithm 4 to reach the Pareto frontier of regret in the unknown Lipschitz setting as described in [11] may be an interesting direction for future research.

The second direction concerns so-called "quantile bounds" [9; 22; 23; 31] for experts setting, where the learner's decision set $\mathcal{W}$ is the simplex $\Delta_d$ and losses are bounded in $\ell_\infty$. The multi-scale machinery developed in the present work is not needed to obtain bounds for this setting because the losses are uniformly bounded across all model classes. Indeed, [14] recovered a basic form of quantile bound using the vanilla multiplicative weights strategy as a meta-algorithm. It is not known whether the more sophisticated data-dependent quantile bounds given in [22; 23] can be recovered in the same fashion.

**Losses with curvature.** The $O(\sqrt{n})$-type regret bounds provided by Algorithm 3 are appropriate when the sub-algorithms themselves incur $O(\sqrt{n})$ regret bounds. However, assuming certain curvature properties (such as strong convexity, exp-concavity, stochastic mixability, etc. [16; 38]) of the loss functions it is possible to construct sub-algorithms that admit significantly more favorable regret bounds ($O(\log n)$ or even $O(1)$). These are also referred to as "fast rates" in online learning. A natural direction for further study is to design a meta-algorithm that admits logarithmic or constant

regret to each sub-algorithm, assuming that the loss functions of interest satisfy similar curvature properties, with the regret to each individual sub-algorithm adapted to the curvature parameters for that sub-algorithm. Perhaps surprisingly, for the special case of the logistic loss, improper prediction and aggregation strategies similar to those proposed in this paper offer a way to circumvent known proper learning lower bounds [17]. This approach will be explored in detail in a forthcoming companion paper.

**Computational efficiency.** We suspect that a running-time of $O(n)$ to obtain inequalities like (6) may be unavoidable through our approach, since we do not make use of the relationship between sub-algorithms beyond using the nested class structure. Whether the runtime of MULTISCALEFTPL can be brought down to match $O(n)$ is an open question. This boils down to whether or not the min-max optimization problem in the algorithm description can simultaneously be solved in 1) Linear time in the number of experts 2) strongly polynomial time in the scales $c_i$.

## Acknowledgements

We thank Francesco Orabona and Dávid Pál for inspiring initial discussions. Part of this work was done while DF was an intern at Google Research and while DF and KS were visiting the Simons Institute for the Theory of Computing. DF is supported by the NDSEG fellowship.

## Footnotes

[1] This regret bound holds under expectation over the player's randomization. It is assumed that each $g_t$ is selected before the randomized strategy $p_t$ is revealed, but may adapt to the distribution over $p_t$. In fact, a slightly stronger version of this bound holds, namely $\mathbb{E}\left[ \sum_{t=1}^{n} \langle e_{i_t}, g_t \rangle - \min_{i \in [N]} \left\{ \sum_{t=1}^{n} \langle e_i, g_t \rangle + O\left( c_i \sqrt{n \log(nc_i/\pi_i)} \right) \right\} \right] \leq 0$. A similar strengthening applies to all subsequent bounds.

[2] For notational convenience all Lipschitz bounds are assumed to be at least 1 without loss of generality for the remainder of the paper.

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
