[Supplementary Material]

# A Proofs

## A.1 Multi-scale FTPL algorithm

**Proof of Theorem 1.** Recall that $B(i) = 5c_i\sqrt{n\big(\log(1/\pi_i) + \log(4c_i^2 n)\big)}$. Let $\mathcal{C} = \big\{g \in \mathbb{R}^N \mid |g_i| \le c_i \ \forall i \in [N]\big\}$. For a regret bound of the form $B(i) + K$ to be achievable by a randomized algorithm such as Algorithm 3 we need

$$\mathcal{V}_n \triangleq \left\langle\!\!\!\left\langle \inf_{P_t \in \Delta(\Delta_N)} \sup_{g_t \in \mathcal{C}} \mathbb{E}_{p_t \sim P_t} \mathbb{E}_{i_t \sim p_t} \right\rangle\!\!\!\right\rangle_{t=1}^n \sup_{i \in [N]}\left[\sum_{t=1}^n \langle e_{i_t}, g_t \rangle - \sum_{t=1}^n \langle e_i, g_t \rangle - B(i)\right] \le K,$$

where $\langle\!\langle \star \rangle\!\rangle_{t=1}^n$ denotes interleaving of the operator $\star$ from $t = 1$ to $n$. In the context of Algorithm 3, the distributions $p_t$ above refer to the strategy $p_t(\sigma_{t+1:n})$ selected by the algorithm and $P_t$ refers to the distribution over this strategy induced by sampling the random variables $\sigma_{t+1:n}$. See [14] for a more extensive introduction to this type of minimax analysis for comparator-dependent regret bounds.

We will develop an algorithm to certify this bound for $K = 1$ using the framework of adaptive relaxations proposed by [14]. Define a relaxation $\mathbf{Rel} : \bigcup_{t=0}^n \mathcal{C}^t \to \mathbb{R}$ via

$$\mathbf{Rel}(g_{1:t}) \triangleq \mathbb{E}_{\sigma_{t+1:n} \in \{\pm 1\}^N} \sup_{i \in [N]}\left[-\sum_{s=1}^t \langle e_i, g_s \rangle + 4\sum_{s=t+1}^n \sigma_s[i] c_i - B(i)\right].$$

The proof structure is as follows: We show that playing $p_t$ as suggested by Algorithm 3 with $\mathbf{Rel}$ satisfies the initial condition and admissibility condition for adaptive relaxations from [14], which implies that if we play $p_t$ we will have $\mathbf{Reg}_n(i) \le B(i) + \mathbf{Rel}(\cdot)$. Then as a final step we bound $\mathbf{Rel}(\cdot)$ using a probabilistic maximal inequality, Lemma 2.

**Initial condition** This condition asks that the initial value of the relaxation $\mathbf{Rel}$ upper bound the worst-case value of the negative benchmark minus the bound $B(i)$ (in other words, the inner part of $\mathcal{V}_n$ with the learner's loss removed). This is holds by definition and is trivial to verify:

$$\mathbf{Rel}(g_{1:n}) = \sup_{i \in [N]}\left[-\sum_{t=1}^n \langle e_i, g_t \rangle - B(i)\right].$$

**Admissibility** For this step we must show that the inequality

$$\inf_{P_t \in \Delta(\Delta_N)} \sup_{g_t \in \mathcal{C}} \mathbb{E}_{p_t \sim P_t} \mathbb{E}_{i_t \sim p_t} \left[\langle e_{i_t}, g_t \rangle + \mathbf{Rel}(g_{1:t})\right] \le \mathbf{Rel}(g_{1:t-1})$$

holds for each timestep $t$, and further that the inequality is certified by the strategy of Algorithm 3. We begin by expanding the definition of $\mathbf{Rel}$:

$$\inf_{P_t \in \Delta(\Delta_N)} \sup_{g_t \in \mathcal{C}} \mathbb{E}_{p_t \sim P_t} \mathbb{E}_{i_t \sim p_t} \left[\langle e_{i_t}, g_t \rangle + \mathbf{Rel}(g_{1:t})\right]$$

$$= \inf_{P_t \in \Delta(\Delta_N)} \sup_{g_t \in \mathcal{C}} \mathbb{E}_{p_t \sim P_t} \mathbb{E}_{i_t \sim p_t} \left[\langle e_{i_t}, g_t \rangle + \mathbb{E}_{\sigma_{t+1:n} \in \{\pm 1\}^N} \sup_{i \in [N]}\left[-\sum_{s=1}^t \langle e_i, g_s \rangle + 4\sum_{s=t+1}^n \sigma_s[i] c_i - B(i)\right]\right].$$

Now plug in the randomized strategy given by Algorithm 3, with $\mathbb{E}_{\sigma_{t+1:n} \in \{\pm 1\}^N}$ taking the place of $\mathbb{E}_{p_t \sim P_t}$:

$$\le \sup_{g_t \in \mathcal{C}}\left[\mathbb{E}_{\sigma_{t+1:n} \in \{\pm 1\}^N}\left[\mathbb{E}_{i_t \sim p_t(\sigma_{t+1:n})} \langle e_{i_t}, g_t \rangle\right] + \mathbb{E}_{\sigma_{t+1:n} \in \{\pm 1\}^N} \sup_{i \in [N]}\left[-\sum_{s=1}^t \langle e_i, g_s \rangle + 4\sum_{s=t+1}^n \sigma_s[i] c_i - B(i)\right]\right].$$

Grouping expectations and applying Jensen's inequality:

$$\le \mathbb{E}_{\sigma_{t+1:n} \in \{\pm 1\}^N} \sup_{g_t \in \mathcal{C}}\left[\mathbb{E}_{i_t \sim p_t(\sigma_{t+1:n})} \langle e_{i_t}, g_t \rangle + \sup_{i \in [N]}\left[-\sum_{s=1}^t \langle e_i, g_s \rangle + 4\sum_{s=t+1}^n \sigma_s[i] c_i - B(i)\right]\right].$$

Expanding the definition of $p_t$ (using its optimality in particular):

$$= \mathbb{E}_{\sigma_{t+1:n} \in \{\pm 1\}^N} \inf_{p_t \in \Delta_N} \sup_{g_t \in \mathcal{C}}\left[\langle p_t, g_t \rangle + \sup_{i \in [N]}\left[-\sum_{s=1}^t \langle e_i, g_s \rangle + 4\sum_{s=t+1}^n \sigma_s[i] c_i - B(i)\right]\right].$$

Now apply a somewhat standard sequential symmetrization procedure. Begin by using the minimax theorem to swap the order of $\inf_{p_t}$ and $\sup_{g_t}$. To do so, we allow the $g_t$ player to randomize, and denote their distribution by $Q_t \in \Delta(\mathcal{C})$.

$$= \mathop{\mathbb{E}}_{\sigma_{t+1:n} \in \{\pm 1\}^N} \sup_{Q_t \in \Delta(\mathcal{C})} \inf_{p_t \in \Delta_N} \mathop{\mathbb{E}}_{g_t \sim Q_t} \left[ \langle p_t, g_t \rangle + \sup_{i \in [N]} \left[ -\sum_{s=1}^{t} \langle e_i, g_s \rangle + 4 \sum_{s=t+1}^{n} \sigma_s[i] c_i - B(i) \right] \right].$$

Since the supremum over $i$ does not directly depend on $p_t$, we can rewrite this expression by introducing a (conditionally) IID copy of $g_t$ which we will denote as $g_t'$:

$$= \mathop{\mathbb{E}}_{\sigma_{t+1:n} \in \{\pm 1\}^N} \sup_{Q_t \in \Delta(\mathcal{C})} \mathop{\mathbb{E}}_{g_t \sim Q_t} \left[ \sup_{i \in [N]} \left[ \inf_{p_t \in \Delta_N} \mathop{\mathbb{E}}_{g_t' \sim Q_t} [\langle p_t, g_t' \rangle] - \sum_{s=1}^{t} \langle e_i, g_s \rangle + 4 \sum_{s=t+1}^{n} \sigma_s[i] c_i - B(i) \right] \right].$$

Choosing $p_t$ to match $e_i$:

$$\leq \mathop{\mathbb{E}}_{\sigma_{t+1:n} \in \{\pm 1\}^N} \sup_{Q_t \in \Delta(\mathcal{C})} \mathop{\mathbb{E}}_{g_t \sim Q_t} \sup_{i \in [N]} \left[ \mathop{\mathbb{E}}_{g_t' \sim Q_t} [\langle e_i, g_t' \rangle] - \langle e_i, g_t \rangle - \sum_{s=1}^{t-1} \langle e_i, g_s \rangle + 4 \sum_{s=t+1}^{n} \sigma_s[i] c_i - B(i) \right].$$

Applying Jensen's inequality:

$$\leq \mathop{\mathbb{E}}_{\sigma_{t+1:n} \in \{\pm 1\}^N} \sup_{Q_t \in \Delta(\mathcal{C})} \mathop{\mathbb{E}}_{g_t, g_t' \sim Q_t} \sup_{i \in [N]} \left[ \langle e_i, g_t' \rangle - \langle e_i, g_t \rangle - \sum_{s=1}^{t-1} \langle e_i, g_s \rangle + 4 \sum_{s=t+1}^{n} \sigma_s[i] c_i - B(i) \right].$$

At this point we can introduce a new Rademacher random variable $\epsilon_t$ without changing the distribution of $g_t' - g_t$, thereby not changing the value of the game:

$$= \mathop{\mathbb{E}}_{\sigma_{t+1:n} \in \{\pm 1\}^N} \sup_{Q_t \in \Delta(\mathcal{C})} \mathop{\mathbb{E}}_{\epsilon_t \in \{\pm 1\}} \mathop{\mathbb{E}}_{g_t, g_t' \sim Q_t} \sup_{i \in [N]} \left[ \epsilon_t \langle e_i, g_t' - g_t \rangle - \sum_{s=1}^{t-1} \langle e_i, g_s \rangle + 4 \sum_{s=t+1}^{n} \sigma_s[i] c_i - B(i) \right]$$

$$\leq \mathop{\mathbb{E}}_{\sigma_{t+1:n} \in \{\pm 1\}^N} \sup_{Q_t \in \Delta(\mathcal{C})} \mathop{\mathbb{E}}_{\epsilon_t \in \{\pm 1\}} \mathop{\mathbb{E}}_{g_t, g_t' \sim Q_t} \left\{ \begin{array}{l} \sup_{i \in [N]} \left[ \epsilon_t \langle e_i, g_t' \rangle + \frac{1}{2} \left( -\sum_{s=1}^{t-1} \langle e_i, g_s \rangle + 4 \sum_{s=t+1}^{n} \sigma_s[i] c_i - B(i) \right) \right] \\ + \sup_{i \in [N]} \left[ \epsilon_t \langle e_i, -g_t \rangle + \frac{1}{2} \left( -\sum_{s=1}^{t-1} \langle e_i, g_s \rangle + 4 \sum_{s=t+1}^{n} \sigma_s[i] c_i - B(i) \right) \right] \end{array} \right\}$$

$$= \mathop{\mathbb{E}}_{\sigma_{t+1:n} \in \{\pm 1\}^N} \sup_{Q_t \in \Delta(\mathcal{C})} \mathop{\mathbb{E}}_{\epsilon_t \in \{\pm 1\}} \mathop{\mathbb{E}}_{g_t \sim Q_t} \sup_{i \in [N]} \left[ 2\epsilon_t \langle e_i, g_t \rangle - \sum_{s=1}^{t-1} \langle e_i, g_s \rangle + 4 \sum_{s=t+1}^{n} \sigma_s[i] c_i - B(i) \right]$$

The above expression is now linear in $Q_t$, so it may be replaced with a pure strategy:

$$= \mathop{\mathbb{E}}_{\sigma_{t+1:n} \in \{\pm 1\}^N} \sup_{g_t \in \mathcal{C}} \mathop{\mathbb{E}}_{\epsilon_t \in \{\pm 1\}} \sup_{i \in [N]} \left[ 2\epsilon_t \langle e_i, g_t \rangle - \sum_{s=1}^{t-1} \langle e_i, g_s \rangle + 4 \sum_{s=t+1}^{n} \sigma_s[i] c_i - B(i) \right]$$

This expression is also convex in $g_t$, which means that the supremum will be obtained at a vertex of $\mathcal{C}$:

$$= \mathop{\mathbb{E}}_{\sigma_{t+1:n} \in \{\pm 1\}^N} \sup_{\sigma_t \in \{\pm 1\}^N} \mathop{\mathbb{E}}_{\epsilon_t \in \{\pm 1\}} \sup_{i \in [N]} \left[ 2\epsilon_t \sigma_t[i] c_i - \sum_{s=1}^{t-1} \langle e_i, g_s \rangle + 4 \sum_{s=t+1}^{n} \sigma_s[i] c_i - B(i) \right]$$

Now apply Theorem 10 conditioned on $\sigma_{t+1:n}$, with $w_i = -\sum_{s=1}^{t-1} \langle e_i, g_s \rangle + 4 \sum_{s=t+1}^{n} \sigma_s[i] c_i - B(i)$.

$$\leq \mathop{\mathbb{E}}_{\sigma_{t:n} \in \{\pm 1\}^N} \sup_{i \in [N]} \left[ -\sum_{s=1}^{t-1} \langle e_i, g_s \rangle + 4 \sum_{s=t}^{n} \sigma_s[i] c_i - B(i) \right]$$

$$= \mathbf{Rel}(g_{1:t-1}).$$

**Final value** The final value of the relaxation is

$$\mathbf{Rel}(\cdot) = 2 \mathop{\mathbb{E}}_{\sigma_{1:n} \in \{\pm 1\}^N} \sup_{i \in [N]} \left[ 2 \sum_{t=1}^{n} \sigma_t[i] c_i - 5 c_i \sqrt{n \left( \log(1/\pi_i) + \log(4 c_i^2 n) \right)} \right] \leq 2 \sum_{i \in [N]} \frac{\pi_i}{4 c_i^2 n} \leq 1.$$

To show the first inequality we have applied a maximal inequality, Lemma 2, by recognizing that $\mathbf{Rel}(\cdot)$ is a supremum of a random process. Namely, we can write $\mathbf{Rel}(\cdot)$ in the form

$\mathbb{E}\sup_{i\in[N]}\{X_i - B(i)\}$ with $X_i = 2\sum_{t=1}^{n}\sigma_t[i]c_i$. The standard mgf bound of $\mathbb{E}\,e^{\lambda X} \leq e^{\lambda^2(b-a)^2/8}$ for mean-zero random variables $X$ with $a \leq X \leq b$ [6], along with independence of the Rademacher random variables in $X_i$, implies that $X_i$ enjoys an mgf bound of

$$\mathbb{E}\,e^{\lambda X_i} \leq e^{2c_i^2\lambda^2 n}.$$

So to prove the result it suffices to take $h_i = 4c_i^2 n$ and $p = 2$ in the statement of Lemma 2 and note that $B(i) \geq (2 + 1/p)h_i^{1/p}(\log(h_i) + \log(1/\pi_i))^{1-1/p}$ in the notation of the lemma. The only additional detail to verify is that, since it was assumed that $c_i \geq 1$ for all $i$ and since $n \geq 1$ by definition, the condition $h_i/\pi_i \geq e$ required by Lemma 2 is satisfied.

**Computational efficiency**   We briefly sketch how the min-max optimization problem in the learner's strategy can be computed efficiently. Recall that the optimization problem is

$$\min_{p\in\Delta_N}\sup_{g_t:|g_t[i]|\leq c_i}\left[\langle p, g_t\rangle + \sup_{i\in[N]}\left[-\sum_{s=1}^{t}\langle e_i, g_s\rangle + 4\sum_{s=t+1}^{n}\sigma_s[i]c_i - B(i)\right]\right]$$

$$= \min_{p\in\Delta_N}\sup_{i\in[N]}\sup_{g_t:|g_t[i]|\leq c_i}\left[\langle p, g_t\rangle - \sum_{s=1}^{t}\langle e_i, g_s\rangle + 4\sum_{s=t+1}^{n}\sigma_s[i]c_i - B(i)\right]$$

Let $G_{t-1}(i) = \sum_{s=1}^{t-1}g_s[i]$. Since the quantity in the brackets above is linear in $g_t$ and there are no interactions between coordinates, we can verify that conditioned on $i$ the max over $g_t$ is obtained via

$$= \min_{p\in\Delta_N}\sup_{i\in[N]}\left[\langle p, c\rangle + (1 - 2p[i])c_i - G_{t-1}(i) + 4\sum_{s=t+1}^{n}\sigma_s[i]c_i - B(i)\right]$$

$$= \min_{p\in\Delta_N}\sup_{i\in[N]}\left[\langle p, c\rangle + \langle a, e_i\rangle - 2\langle p, \mathrm{diag}(c)e_i\rangle\right],$$

where $a[i] = c_i - G_{t-1}(i) + 4\sum_{s=t+1}^{n}\sigma_s[i]c_i - B(i)$. We can now employ a standard reduction from saddle point optimization to linear programming, i.e.

$$\begin{aligned}\text{minimize}\quad &\langle p, c\rangle + s\\ \text{subject to}\quad &s \geq \langle a, e_i\rangle - 2\langle p, \mathrm{diag}(c)e_i\rangle \quad \forall i.\\ &p \in \Delta_N.\end{aligned}$$

Assuming that $\min_i c_i \geq 1$, this linear program can be solved to accuracy $\epsilon$ by interior point methods (e.g. [35]) in time $O(N^{3.5}\log(\epsilon^{-1}\max_i c_i))$ or by Mirror-Prox [28] in time $O(N\epsilon^{-1}\max_i c_i)$. Since our rates scale as $\sqrt{n}$ we can set $\epsilon = 1/(\sqrt{n}\max_i c_i)$ to conclude the result.

As a final implementation detail, we remark that similar to the FTPL algorithm in [34] one can draw each perturbation $\sigma_t[i]$, from the distribution $\mathcal{N}(0, 1)$ instead of using Rademacher random variables. This allows one to replace each sum $\sum_{s=t}^{n}\sigma_s[i]$ with a draw from $\mathcal{N}(0, n-t)$ and therefore avoid spending $O(n)$ time per step sampling perturbations. We have omitted the details because — for most values of $c$ and $N$ used in our applications, at least — the time required to solve the saddle point optimization problem dominates the runtime, not the time to sample perturbations.

□

**Theorem 10.** *For any $w \in \mathbb{R}^N$, any $c \in \mathbb{R}_+^N$,*

$$\sup_{\sigma\in\{\pm 1\}^N}\mathbb{E}_{\epsilon\in\{\pm 1\}^N}\max_{i\in[N]}\{w_i + 2\epsilon\sigma_i c_i\} \leq \mathbb{E}_{\sigma\in\{\pm 1\}^N}\max_{i\in[N]}\{w_i + 4\sigma_i c_i\}. \tag{11}$$

**Proof of Theorem 10.** Fix any $\sigma \in \{\pm 1\}^N$. Let $i_1 = \arg\max_{i\in[N]}\{w_i + 2\sigma_i c_i\}$ and $i_{-1} = \arg\max_{i\in[N]}\{w_i - 2\sigma_i c_i\}$. Then it is easy to see that

$$\mathbb{E}_{\epsilon}\max_{i\in[N]}\{w_i + 2\epsilon\sigma_i c_i\} = \mathbb{E}_{\epsilon}\max_{i\in\{i_1, i_{-1}\}}\{w_i + 2\epsilon\sigma_i c_i\} \leq \mathbb{E}_{\sigma'\in\{\pm 1\}^N}\max_{i\in\{i_1, i_{-1}\}}\{w_i + 4\sigma_i' c_i\} \leq \mathbb{E}_{\sigma'\in\{\pm 1\}^N}\max_{i\in[N]}\{w_i + 4\sigma_i' c_i\}.$$

The central inequality above follows by Lemma 1 with the pair $(w, 2c)$. Since the above bound holds for any $\sigma$, we conclude that (11) holds.

□

**Lemma 1.** For any pair $(w, c)$ where $w \in \mathbb{R}^N$ any $c \in \mathbb{R}_+^N$, the inequality

$$\sup_{\sigma \in \{\pm 1\}^N} \mathbb{E}_{\epsilon \in \{\pm 1\}} \max_{i \in [N]} \{w_i + \epsilon \sigma_i c_i\} \leq \mathbb{E}_{\sigma \in \{\pm 1\}^N} \max_{i \in [N]} \{w_i + 2\sigma_i c_i\}. \tag{12}$$

holds when $N = 2$.

**Proof of Lemma 1.** In this proof we adopt the notation that for any element $j \in [2]$, $-j$ denote the other element. Say the pair $(w, c)$ is *dominated* if there exists $j$ for which $w_j - c_j \geq w_{-j} + c_{-j}$. Note that this of course implies $w_j + c_j \geq w_{-j} + c_{-j}$ as well, since $c$ is non-negative.

**Dominated case** Suppose $(w, c)$ is dominated by index $j$. Then (12) holds trivially for any $K \in \mathbb{R}$ by

$$\sup_{\sigma \in \{\pm 1\}^N} \mathbb{E}_{\epsilon \in \{\pm 1\}} \max_{i \in [N]} \{w_i + \epsilon \sigma_i c_i\} = w_j = \max_{i \in [N]} \{w_i + K \mathbb{E}_{\sigma \in \{\pm 1\}^N} \sigma_i c_i\} \leq \mathbb{E}_{\sigma \in \{\pm 1\}^N} \max_{i \in [N]} \{w_i + K \sigma_i c_i\}.$$

We now focus on the trickier "not dominated" case.

**Rescaling doesn't induce domination** We first observe that if $(w, c)$ does is not dominated, $(w, Bc)$ is not dominated either for any $B \geq 1$. Let $j$ be the index for which $w_j + c_j \geq w_{-j} + c_{-j}$ which implies $w_j - c_j \leq w_{-j} + c_{-j}$ because $(w, c)$ is not dominated. Observe that if $(w, Bc)$ is dominated we either have $w_j - Bc_j \geq w_{-j} + Bc_{-j}$ or $w_{-j} - Bc_{-j} \geq w_j + Bc_j$. The first case cannot hold because $B \geq 1$ and we already know that $(w, c)$ is not dominated. The second case in particular implies $w_{-j} \geq w_j$, so we must have had $c_j \geq c_{-j}$ to begin with. But in that case we will still have $w_j + Bc_j \geq w_{-j} + Bc_{-j}$ which contradicts the domination.

Note: It is good to keep in mind that while rescaling does not induce domination, it may not be the case in general that $w_j + Bc_j \geq w_{-j} + Bc_{-j}$ even though $w_j + c_j \geq w_{-j} + c_{-j}$. That is, the "leader" may change after rescaling.

**LHS of (12) for $(w, c)$ not dominated** When $(w, c)$ is not dominated we have

$$\sup_{\sigma \in \{\pm 1\}^N} \mathbb{E}_{\epsilon \in \{\pm 1\}} \max_{i \in [N]} \{w_i + \epsilon \sigma_i c_i\} = \frac{1}{2}(w_1 + c_1) + \frac{1}{2}(w_2 + c_2).$$

**RHS of (12) for $(w, c)$ not dominated** We will consider the RHS of (12) for $(w, c') \triangleq (w, Bc)$ for some $B \geq 1$ to be decided. By the argument above, the pair $(w, c')$ is also not dominated. For the remainder of the proof, 1 will denote the index for which $w_1 + c_1' \geq w_2 + c_2'$. Because the pair is not dominated, the value the RHS takes can be classified into two cases based on the relationship between $c'$ and $w$.

- Case 1: $w_1 - c_1' \leq w_2 - c_2'$:
  In this case there is equal probability that the process takes on value $w_2 - c_2'$ or $w_2 + c_2'$ conditioned on the event that $\sigma_1 = -1$, so we have the equality:

  $$\mathbb{E}_{\sigma \in \{\pm 1\}^N} \max_{i \in [N]} \{w_i + \sigma_i c_i'\} = \frac{1}{2}(w_1 + w_2) + \frac{1}{2}c_1'$$

  Furthermore, Case 1 implies $c_1' \geq c_2'$, which leads to an inequality:

  $$\geq \frac{1}{2}(w_1 + w_2) + \frac{1}{4}(c_1' + c_2').$$

- Case 2: $w_1 - c_1' \geq w_2 - c_2'$:
  In this case, conditioned on the event that $\sigma_1 = -1$, there is equal probability that the process takes on value $w_2 + c_2'$ or $w_1 - c_1'$, so the equality becomes:

  $$\mathbb{E}_{\sigma \in \{\pm 1\}^N} \max_{i \in [N]} \{w_i + \sigma_i c_i'\} = \frac{1}{2}(w_1 + c_1') + \frac{1}{4}(w_2 + c_2') + \frac{1}{4}(w_1 - c_1')$$

  Case 2 implies that $w_1 \geq w_2$, because we may add the inequalities $w_1 + c_1' \geq w_2 + c_2'$ and $w_1 - c_1' \geq w_2 - c_2'$. This gives an inequality:

  $$\geq \frac{1}{2}(w_1 + w_2) + \frac{1}{4}(c_1' + c_2').$$

Combining our results for the two cases, we have that for any vector $c'$, so long as $(w, c')$ is not dominated,

$$\mathbb{E}_{\sigma \in \{\pm 1\}^N} \max_{i \in [N]} \{w_i + \sigma_i c'_i\} \geq \frac{1}{2}(w_1 + w_2) + \frac{1}{4}(c'_1 + c'_2).$$

In particular, choosing $B = 2$ implies (12) in the non-dominated case:

$$\mathbb{E}_{\sigma \in \{\pm 1\}^N} \max_{i \in [N]} \{w_i + 2\sigma_i c_i\} \geq \frac{1}{2}(w_1 + w_2) + \frac{1}{2}(c_1 + c_2)$$
$$= \sup_{\sigma \in \{\pm 1\}^N} \mathbb{E}_\epsilon \max_{i \in [N]} \{w_i + \epsilon \sigma_i c_i\}.$$

**Final result** Combining the dominated and non-dominated results we have that for any $(w, c)$.

$$\sup_{\sigma \in \{\pm 1\}^N} \mathbb{E}_\epsilon \max_{i \in [N]} \{w_i + \epsilon \sigma_i c_i\} \leq \mathbb{E}_{\sigma \in \{\pm 1\}^N} \max_{i \in [N]} \{w_i + 2\sigma_i c_i\}.$$

□

**Lemma 2** (Multi-scale maximal inequality). Let $(X_i)_{i \in [N]}$ be a real-valued random process for which there exists a sequence $(h_i)_{i \in [N]}$ with $h_i > 0$ such that the moment generating function bound $\mathbb{E}\,e^{\lambda X_i} \leq e^{\lambda^p h_i}$ is satisfied for all $\lambda > 0$ and some choice of $p > 0$. Then for any distribution $\pi \in \Delta_N$ for which $h_i/\pi_i \geq e$ for all $i \in [N]$ it holds that

$$\mathbb{E} \sup_{i \in [N]} \left\{ X_i - (2 + 1/p)h_i^{1/p}(\log(h_i) + \log(1/\pi_i))^{1-1/p} \right\} \leq \sum_{i \in [N]} \frac{\pi_i}{h_i}. \qquad (13)$$

**Proof.** Let $B(i) = Ch_i^{1/p}(\log(h_i) + \log(1/\pi_i))^{1-1/p}$ for some constant $C$ to be decided later. One should verify that $\log(h_i) + \log(1/\pi_i)$ is always non-negative by the assumption that $h_i/\pi_i \geq e$, which will be used repeatedly. To begin, observe that

$$\mathbb{E} \sup_{i \in [N]} \{X_i - B(i)\} \leq \mathbb{E} \sup_{i \in [N]} [X_i - B(i)]_+,$$

where $[x]_+ = \max\{x, 0\}$. By non-negativity of $[x]_+$ it further holds that

$$\leq \mathbb{E} \sum_{i \in [N]} [X_i - B(i)]_+.$$

Fixing an arbitrary sequence $(\lambda_i)_{i \in [N]}$ with $\lambda_i > 0$, the basic inequality $\max\{a, b\} \leq \frac{1}{\lambda}\log(e^{\lambda a} + e^{\lambda b})$ implies the following upper bound:

$$\leq \mathbb{E} \sum_{i \in [N]} \frac{1}{\lambda_i} \log\left(1 + e^{\lambda_i(X_i - B(i))}\right).$$

Apply Jensen's inequality:

$$\leq \sum_{i \in [N]} \frac{1}{\lambda_i} \log\left(1 + \mathbb{E}\,e^{\lambda_i(X_i - B(i))}\right).$$

Now use the moment bound assumed in the lemma statement:

$$\leq \sum_{i \in [N]} \frac{1}{\lambda_i} \log\left(1 + e^{\left(\lambda_i^p h_i - \lambda_i B(i)\right)}\right).$$

Lastly, apply the inequality $\log(1 + x) \leq x$ for $x \geq 0$:

$$\leq \sum_{i \in [N]} \exp\left(\lambda_i^p h_i - \lambda_i B(i) + \log(1/\lambda_i)\right).$$

We now take $\lambda_i = \left(\frac{\log(h_i) + \log(1/\pi_i)}{h_i}\right)^{1/p}$ and bound each exponent in the sum above. Using the definition of $B(i)$:

$$\lambda_i^p h_i - \lambda_i B(i) + \log(1/\lambda_i) = \log(1/\lambda_i) - (C - 1)(\log(1/\pi_i) + \log(h_i)).$$

Next observe that

$$\log(1/\lambda_i) = \frac{1}{p}\log\left(\frac{h_i}{\log(h_i/\pi_i)}\right) \le \frac{1}{p}\log(h_i),$$

where we have used that $h_i/\pi_i \ge e$. With this, and using that $\log(1/\pi_i) \ge 0$, we have

$$\lambda_i^p h_i - \lambda_i B(i) + \log(1/\lambda_i) \le -(C - 1 - 1/p)(\log(1/\pi_i) + \log(h_i)).$$

Taking $C \ge 2 + 1/p$ and using this bound in the summation over $i$ yields the result:

$$\mathbb{E}\sup_{i\in[N]}\{X_i - B(i)\} \le \sum_{i\in[N]}\frac{\pi_i}{h_i}.$$

$\square$

## A.2 Proofs for Section 2.2

**Proof of Theorem 2.** First, we verify that the loss sequence $(g_t)_{t\le n}$ is such that the regret bound derived for MULTISCALEFTPL applies. In particular, we need to verify that $|g_t[i]| \le c_i$ for each $i$. To this end, fix an index $i \in [N]$, and note that since $f_t$ is $L_i$-Lipschitz on $\mathcal{W}_i$ with respect to the norm $\|\cdot\|_{(i)}$ we have

$$|g_t[i]| = |f_t(w_t^i) - f_t(0)| \le L_i\|w_t^i - 0\|_{(i)} \le L_i R_i \le L_i R_i = c_i,$$

as required. Also, it was assumed that $c_i = L_i R_i \ge 1$, as required for Theorem 1.

Now, recall that $(p_t)$ is the sequence of distributions produced by the meta-algorithm. The algorithm's total loss with respect to the centered iterates $(\widetilde{f}_t)$ is given by

$$\sum_{t=1}^{n}\widetilde{f}_t(w_t^{i_t}) = \sum_{t=1}^{n}\langle e_{i_t}, g_t\rangle,$$

where this equality is due to the construction of the losses $(g_t)_{t\le n}$ given to MULTISCALEFTPL. The regret bound for MULTISCALEFTPL now implies that

$$\mathbb{E}\left[\sum_{t=1}^{n}\langle e_{i_t}, g_t\rangle - \min_{i\in[N]}\left\{\sum_{t=1}^{n}g_t[i] + O\left(R_i L_i\sqrt{n\log(R_i L_i n/\pi_i)}\right)\right\}\right] \le 0,$$

where we have obtained this inequality by substituting the value of the vector $c$ constructed by MULTISCALEOCO into the regret bound (4) for MULTISCALEFTPL. Now, observe that for each $i$ we have

$$\sum_{t=1}^{n}g_t[i] = \sum_{t=1}^{n}\widetilde{f}_t(w_t^i) \le \inf_{w\in\mathcal{W}_i}\sum_{t=1}^{n}\widetilde{f}_t(w) + \mathbf{Reg}_n(i),$$

where we have used the definition of $g_t$ and the regret bound assumed on the sub-algorithm. Combining these inequalities, we have

$$\mathbb{E}\left[\sum_{t=1}^{n}\widetilde{f}_t(w_t^{i_t}) - \min_{i\in[N]}\left\{\inf_{w\in\mathcal{W}_i}\sum_{t=1}^{n}\widetilde{f}_t(w) + \mathbf{Reg}_n(i) + O\left(R_i L_i\sqrt{n\log(R_i L_i n/\pi_i)}\right)\right\}\right] \le 0.$$

Finally, observe that since $\widetilde{f}_t(w) = f_t(w) - f_t(0)$, the above is equivalent to

$$\mathbb{E}\left[\sum_{t=1}^{n}f_t(w_t^{i_t}) - \min_{i\in[N]}\left\{\inf_{w\in\mathcal{W}_i}\sum_{t=1}^{n}f_t(w) + \mathbf{Reg}_n(i) + O\left(R_i L_i\sqrt{n\log(R_i L_i n/\pi_i)}\right)\right\}\right] \le 0.$$

$\square$

**Mirror Descent**  Online Mirror Descent is the standard algorithm for online linear optimization over convex sets. It is parameterized by a convex set $\mathcal{W}$, learning rate $\eta$, and strongly convex regularizer $\mathcal{R} : \mathcal{W} \to \mathbb{R}$. We define the update $\textsc{MirrorDescent}(\eta, \mathcal{W}, \mathcal{R})$ as follows. First, set $w_1 = \arg\min_{w \in \mathcal{W}} \mathcal{R}(w)$. Then, for each time $t \in [n]$:

- Receive gradient $g_t$ and let $\widetilde{w}_{t+1}$ satisfy $\nabla \mathcal{R}(\widetilde{w}_{t+1}) = \nabla \mathcal{R}(w_t) - \eta g_t$.

- Set $w_{t+1} = \arg\min_{w \in \mathcal{W}} \mathcal{D}_{\mathcal{R}}(w \mid \widetilde{w}_{t+1})$.

**Fact 1** (Mirror Descent (e.g. [15]))**.** Let $(w_t)$ be the iterates produced by $\textsc{MirrorDescent}(\eta, \mathcal{W}, \mathcal{R})$ on a sequence of vectors $(g_t)_{t \leq n}$. If $\mathcal{R}$ is $\lambda$-strongly convex with respect to a norm $\|\cdot\|_{\mathcal{R}}$, the iterates satisfy

$$\sum_{t=1}^{n} \langle w_t - w, g_t \rangle \leq \frac{\eta}{2\lambda} \sum_{t=1}^{n} \|g_t\|_{\mathcal{R},\star}^2 + \frac{1}{\eta} \mathcal{R}(w) \quad \forall w \in \mathcal{W}. \tag{14}$$

**Proof of Theorem 3.** Recall that each sub-algorithm $\textsc{Alg}_i$ runs Mirror Descent over a ball in $(\mathfrak{B}, \|\cdot\|)$ of radius $R_i$ using the regularizer $\mathcal{R}(w) = \frac{1}{2}\|w\|^2$. From the regret bound for Mirror Descent (Fact 1), the meta-algorithm's choice of Mirror Descent parameters for $\textsc{Alg}_i$ (in particular, the choice $\eta_i = \frac{R_i}{L}\sqrt{\frac{\lambda}{n}}$) guarantees that

$$\sum_{t=1}^{n} f_t(w_t^i) - \inf_{w \in \mathcal{W}_i} \sum_{t=1}^{n} f_t(w) \leq O(R_i L \sqrt{n/\lambda}).$$

Combined with the regret bound for $\textsc{MultiScaleOCO}$ (Theorem 2, noting that $R_i L_i = R_i L \geq 1$), this implies that the meta-algorithm's regret satisfies

$$\mathbb{E}\left[\sum_{t=1}^{n} f_t(w_t^{i_t}) - \min_{i \in [N]}\left\{\inf_{w \in \mathcal{W}_i} \sum_{t=1}^{n} f_t(w) + O(R_i L\sqrt{n/\lambda}) + O\left(R_i L\sqrt{n\log(R_i L n/\pi_i)}\right)\right\}\right] \leq 0.$$

Which, using that $\pi_i = 1/(n+1)$ and combining terms, further implies

$$\mathbb{E}\left[\sum_{t=1}^{n} f_t(w_t^{i_t}) - \min_{i \in [N]}\left\{\inf_{w \in \mathcal{W}_i} \sum_{t=1}^{n} f_t(w) + O\left(R_i L\sqrt{n\log(R_i L n)/\lambda}\right)\right\}\right] \leq 0.$$

Now, recall that $i \in [n+1]$, and that $R_i = e^{i-1}$. Consider the algorithm's regret against a comparator $w$. For now, assume that $w$ satisfies $1 \leq \|w\| \leq e^n$ — we will see shortly that this is without loss of generality. Let $i^\star(w) = \min\{i \mid w \in \mathcal{W}_i\}$. Then the regret bound above implies

$$\mathbb{E}\left[\sum_{t=1}^{n} f_t(w_t^{i_t}) - \left\{\sum_{t=1}^{n} f_t(w) + O\left(R_{i^\star(w)} L\sqrt{n\log\left(R_{i^\star(w)} L n\right)/\lambda}\right)\right\}\right] \leq 0.$$

Furthermore, since $R_i = e^{i-1}$, we have that $R_{i^\star(w)} \leq e\|w\|$, and so

$$\mathbb{E}\left[\sum_{t=1}^{n} f_t(w_t^{i_t}) - \left\{\sum_{t=1}^{n} f_t(w) + O\left(\|w\| L\sqrt{n\log(\|w\| L n/)\lambda}\right)\right\}\right] \leq 0.$$

This is exactly the regret bound we wanted. Now, the case where $\|w\| \leq 1$ is handled by simply noting $i^\star(w) = 1$ and writing $R_1 = 1 \leq 1 + \|w\|$, which gives the $\|w\| + 1$ factor as follows:

$$\mathbb{E}\left[\sum_{t=1}^{n} f_t(w_t^{i_t}) - \left\{\sum_{t=1}^{n} f_t(w) + O\left((\|w\| + 1) L\sqrt{n\log((\|w\| + 1) L n/)\lambda}\right)\right\}\right] \leq 0.$$

To handle the case where $\|w\| \geq e^n$ we appeal to Corollary 1 with $c = L\sqrt{n}$ and $\gamma = 1/2$, which shows that it suffices to consider only $\|w\| \leq \exp\left(\left(\frac{Ln}{c}\right)^{1/\gamma}\right) = e^n$. Note that the constants appearing in the regret bound above, both inside the $O(\cdot)$ and inside the $\sqrt{\log(\cdot)}$ are worse than those with which we instantiate Corollary 1. This is not an issue because worse constants only reduce the radius that must be considered in the corollary. $\qquad\square$

**Lemma 3.** Let $F : \mathbb{R}_+ \to \mathbb{R}_+$ be given. Suppose the loss sequence $(f_t)_{t \le n}$ is $L$-Lipschitz with respect to $\|\cdot\|_\star$. Then a regret bound of the form

$$\sum_{t=1}^n f_t(w_t) - \sum_{t=1}^n f_t(w) \le F(\|w\|) \quad \forall w \in \mathfrak{B} \tag{15}$$

holds if the restricted regret bound

$$\sum_{t=1}^n f_t(w_t) - \sum_{t=1}^n f_t(w) \le F(\|w\|) \quad \forall f : \|f\| \le \alpha^\star, \tag{16}$$

holds, where $\alpha^\star$ is the greatest non-negative number for which $F(\alpha^\star) - \alpha^\star Ln \ge F(0)$.

**Proof of Lemma 3.** Assume wlog that $f_t(0) = 0$ for each $t$. This is possible because

$$\sum_{t=1}^n f_t(w_t) - \sum_{t=1}^n f_t(w) = \sum_{t=1}^n (f_t(w_t) - f_t(0)) - \sum_{t=1}^n (f_t(w) - f_t(0)).$$

To begin, observe that (15) is equivalent to

$$\sum_{t=1}^n f_t(w_t) \le \inf_{w \in \mathfrak{B}} \left\{ \sum_{t=1}^n f_t(w) + F(\|w\|) \right\}.$$

By selecting $w = 0$, $f_t(0) = 0$ implies that the infimum on the right is always upper bounded in value by $F(0)$. In the other direction, Lipschitzness of the losses along with $f_t(0) = 0$ implies that the infimum is lower bounded as

$$\inf_{w \in \mathfrak{B}} \left\{ \sum_{t=1}^n f_t(w) + F(\|w\|) \right\} \ge \inf_{w \in \mathfrak{B}} \{ -L\|w\|n + F(\|w\|) \} = \inf_{\alpha \ge 0} \{ -\alpha Ln + F(\alpha) \}.$$

Therefore if $\alpha \ge \alpha^\star$, the lower bound $-\alpha Ln + F(\alpha)$ will be sub-optimal compared to the upper bound of $F(0)$ obtained by choosing $\alpha = 0$. $\qquad\square$

**Corollary 1.** When $F(r) = c \cdot (r+1) \log(r+1)^\gamma$ for $\gamma > 0$, it is sufficient to consider

$$\sum_{t=1}^n f_t(w_t) - \sum_{t=1}^n f_t(w) \le F(\|w\|) \quad \forall w : \|w\| \le \exp\left( \left( \frac{Ln}{c} \right)^{1/\gamma} \right). \tag{17}$$

**Proof of Corollary 1.** Note that $F(0) = 0$. Let $r$ denote the minimizer of $F(\alpha) - \alpha \cdot a$ (where $a = Ln$). Differentiating this expression yields

$$a = c\left( \log(r+1)^\gamma + \gamma \log(r+1)^{\gamma-1} \right),$$

which further implies

$$\log(r+1)^\gamma = \frac{a}{c} \cdot \frac{1}{1 + \gamma/\log(r+1)} \le \frac{a}{c}.$$

Rearranging, we have $r \le \exp((a/c)^{1/\gamma}) - 1$. Since $F(\alpha) - \alpha \cdot a$ is strictly convex, this function is increasing above $r$. To conclude, we guess an upper bound on the value of $\alpha^\star$: $\alpha \coloneqq \exp((a/c)^{1/\gamma}) - 1$. Substituting this value in, we have

$$F(\alpha) - \alpha \cdot a \ge a \exp((a/c)^{1/\gamma}) - a \cdot \exp((a/c)^{1/\gamma}) = 0 = F(0),$$

which yields the result. $\qquad\square$

**Proof of Theorem 4.** We only sketch the details of this proof as it follows Theorem 3 very closely.

We first describe sub-algorithm configuration for MULTISCALEOCO that achieves the claimed regret bound. Our strategy will be to take a discretization the range of $p$ values $[1 + \delta, 2]$, and produce a set of sub-algorithms for each $p$ in this discrete set. For a fixed $p$, the construction of the set of sub-algorithms will be exactly is in Theorem 3. The discrete set of $p$s will have the form $p_k = 1 + \delta + \min\{(k-1) \cdot \epsilon, (1-\delta)\}$, for $\epsilon = 1/\log(d)$ and $k \in [1, \ldots, K]$, where $K = \lceil (1-\delta)/\epsilon \rceil + 1$ (in particular $k \le \log(d) + 1$).

For a fixed $k$, the norm $\|\cdot\|_{p_k}$ has that $\frac{1}{2}\|\cdot\|_{p_k}^2$ is $(p_k - 1)$-strongly convex with respect to itself [19]. With this in mind, we create a set of $N \coloneqq K(n+1)$ sub-algorithms, which we will index by pairs $(k, j) \in [K] \times [n+1]$ instead of $i \in [K(n+1)]$ for notational convenience.

- For each $k \in [K]$:
  - $L_k = L_{p_k}$.
  - For each $j \in \{1, \ldots, n+1\}$:
    * Set $R_j = e^{j-1}$.
    * Take $\mathcal{W}_{(k,j)} = \{w \in \mathfrak{B} \mid \|w\|_{p_k} \le R_j\}$, $\eta_{(k,j)} = \frac{R_j}{L_k}\sqrt{\frac{\lambda_{p_k}}{n}}$, where $\lambda_{p_k} = (p_k - 1)$.
    * Let $\text{ALG}_j = \text{MIRRORDESCENT}(\eta_{(k,j)}, \mathcal{W}_{(k,j)}, \|\cdot\|_{p_k}^2)$.
- $\pi = \text{Uniform}([K] \times [n+1])$.

Clearly the total number of sub-algorithms and hence the running time scales as $O(n \cdot \log(d))$.

Referring back to the proof of Theorem 3, and letting $(k_t, j_t)$ denote the index pair chosen by MULTISCALEOCO in round $t$, it is clear that for a fixed $k$, the algorithm satisfies for all $w \in \mathbb{R}^d$

$$\mathbb{E}\left[\sum_{t=1}^n f_t(w_t^{(k_t, j_t)}) - \left\{\sum_{t=1}^n f_t(w) + O\left((\|w\|_{p_k} + 1)L_{p_k}\sqrt{n\log((\|w\|_{p_k} + 1)L_{p_k} n \log(d))/(p_k - 1)}\right)\right\}\right] \le 0.$$

In fact, the regret guarantee for MULTISCALEOCO implies that

$$\mathbb{E}\left[\sum_{t=1}^n f_t(w_t^{(k_t, j_t)}) - \min_{k \in [N]}\left\{\sum_{t=1}^n f_t(w) + O\left((\|w\|_{p_k} + 1)L_{p_k}\sqrt{n\log((\|w\|_{p_k} + 1)L_{p_k} n \log(d))/(p_k - 1)}\right)\right\}\right] \le 0. \tag{18}$$

We now appeal to the choice of discretization to deduce that

$$\mathbb{E}\left[\sum_{t=1}^n f_t(w_t^{(k_t, j_t)}) - \min_{p \in [1+\delta, 2]}\left\{\sum_{t=1}^n f_t(w) + O\left((\|w\|_p + 1)L_p\sqrt{n\log((\|w\|_p + 1)L_p \log(d)n)/(p - 1)}\right)\right\}\right] \le 0.$$

Suppose there is some $p \in [1+\delta, 2]$ of interest. Let $k$ be the greatest integer for which $p_k \le p$. We claim that the bound

$$\mathbb{E}\left[\sum_{t=1}^n f_t(w_t^{(k_t, j_t)}) - \left\{\sum_{t=1}^n f_t(w) + O\left((\|w\|_{p_k} + 1)L_{p_k}\sqrt{n\log((\|w\|_{p_k} + 1)L_{p_k} n \log(d))/(p_k - 1)}\right)\right\}\right] \le 0,$$

implies the desired result. By duality we have that $\|w\|_{p_k} \ge \|w\|_p$ and $L_{p_k} \le L_p$. To conclude, observe that $\|w\|_{p_k}/\|w\|_p \le \|w\|_{p_k}/\|w\|_{p_{k+1}} \le d^\epsilon = d^{1/\log(d)} = O(1)$, so the norm terms in the bound above are within constant factors of the desired bound. $\qquad\square$

**Proof of Theorem 5.** Recall that for fixed $k$, the learner predicts from a class

$$\mathcal{W}_k = \{W \in \mathbb{R}^{d \times d} \mid W \succeq 0, \|W\|_\sigma \le 1, \langle W, I\rangle = k\},$$

and experiences affine losses $f_t(W_t) = \langle I - W_t, Y_t\rangle$, where $Y_t \in \mathcal{Y} := \{Y \in \mathbb{R}^{d \times d} \mid Y \succeq 0, \|Y\|_\sigma \le 1\}$. The regret for this game is given by

$$\sup_{W \in \mathcal{W}_k}\left[\sum_{t=1}^n \langle I - W_t, Y_t\rangle - \sum_{t=1}^n \langle I - W, Y_t\rangle\right]. \tag{19}$$

From [29], we have that for fixed $k$ the strategy MATRIX EXPONENTIATED GRADIENT has regret bounded by

$$O\left(\min\left\{\sqrt{nk^2\log(n/k)}, \sqrt{n(d-k)^2\log(n/(d-k))}\right\}\right) = \widetilde{O}\left(\sqrt{n\min\{k, d-k\}^2}\right).$$

Note: The variant of MATRIX EXPONENTIATED GRADIENT that obtains this strategy uses either losses or gains depending on the value of $k$. See [29] for more details.

The configuration with which we invoke MULTISCALEOCO is:

- For each $i \in [\lceil \log(d/2)\rceil + 1]$:
  - Set $R_i = e^{i-1}$, $L_i = 1$.

- $\mathcal{W}_i = \left\{ W \in \mathbb{R}^{d \times d} \mid W \succeq 0, \|W\|_\sigma \leq 1, \langle W, I \rangle = R_i \right\}$

  - Take $\text{ALG}_i = \text{MATRIX EXPONENTIATED GRADIENT}(\mathcal{W}_i)$ as described in [29].

- $\pi = \text{Uniform}([\lceil \log(d/2) \rceil + 1])$.

As in Theorem 3 and Theorem 4, choosing $R_i$ to be spaced exponentially is sufficient to guarantee that there is a sub-algorithm whose regret is within a constant factor $e$ of $\widetilde{O}\left(k\sqrt{n}\right)$ for any choice of the rank $k$.

All that remains is that the losses of the sub-algorithms satisfy the claimed upper bound $R_i$. Observe that MULTISCALEOCO works with centered loss $\widetilde{f}_t(W) = -\langle W, Y_t \rangle$. For any $W \in \mathcal{W}_k$, we have

$$|\langle W, Y_t \rangle| \leq \|Y_t\|_\sigma \|W\|_\Sigma \leq 1 \cdot R_k,$$

so the condition is satisfied. $\qquad\square$

**Proof of Theorem 6.** We will use a meta-algorithm strategy closely resembling that of the smooth Banach space setting. The only difference is that $\|\cdot\|_\Sigma$ is not smooth, so MATRIX MULTIPLICATIVE WEIGHTS, which uses the log-trace-exponential function as a surrogate for $\|\cdot\|_\Sigma$, is used as the sub-algorithm instead of working with $\|\cdot\|_\Sigma$ directly.

We use the version of MATRIX MULTIPLICATIVE WEIGHTS stated in [18] Theorem 13, which uses classes of the form $\mathcal{W}_r = \left\{ W \in \mathbb{R}^{d \times d} \mid W \succeq 0, \|W\|_\Sigma \leq r \right\}$ and has regret against $\mathcal{W}_r$ bounded by $O(r\sqrt{n \log d})$ whenever each loss matrix $Y_t$ has $\|Y_t\|_\sigma \leq 1$. Using this strategy for fixed $r$ as a sub-algorithm for MULTISCALEOCO, we achieve the following oracle inequality efficiently:

For each $i \in [n+1]$:

- Set $R_i = 2^{i-1}$

- $L_i = 1$ (we are assuming $\|Y_t\|_\sigma \leq 1$).

- $\mathcal{W}_i = \left\{ W \in \mathbb{R}^{d \times d} \mid W \succeq 0, \|W\|_\Sigma \leq R_i \right\}$

- $\text{ALG}_i = \text{MATRIX MULTIPLICATIVE WEIGHTS}(\mathcal{W}_i)$

Finally, we set $\pi = \text{Uniform}([n+1])$. That this configuration is sufficient follows from the doubling analysis given in the proof of Theorem 3. Losses are once again bounded via $|\langle W, Y_t \rangle| \leq \|W\|_\Sigma \|Y_t\|_\sigma \leq R_i$ for $W \in \mathcal{W}_i$. $\qquad\square$

## A.3 Proofs from Section 2.3

---

**Algorithm 5**

---
**procedure** MULTISCALELEARNING($\{\text{ALG}_i, R_i, L_i\}_{i \in [N]}, \pi$)  $\qquad \triangleright$ Collection of sub-algorithms, prior $\pi$.
    $c \leftarrow (R_i \cdot L_i)_{i \in [N]}$  $\qquad\qquad\qquad\qquad\qquad\qquad\quad \triangleright$ Sub-algorithm scale parameters.
    Define $\widetilde{\ell}(\hat{y}, y) = \ell(\hat{y}, y) - \ell(0, y)$.  $\qquad\qquad\qquad\qquad \triangleright$ Center the loss function.
    **for** $t = 1, \ldots, n$ **do**
        Receive context $x_t$
        $\hat{y}_t^i \leftarrow \text{ALG}_i((x_1, y_1), \ldots, (x_{t-1}, y_{t-1}), x_t)$ for each $i \in [N]$.
        $i_t \leftarrow \text{MULTISCALEFTPL}[c, \pi](g_1, \ldots, g_{t-1})$.
        Play $\hat{y}_t = \hat{y}_t^{i_t}$.
        Observe $y_t$ and let $g_t = \left( \widetilde{\ell}_t(\hat{y}_t^i, y_t) \right)_{i \in [N]}$.
    **end for**
**end procedure**

---

**Proof of Theorem 7.** This theorem is an immediate consequence of Theorem 2, using the absolute value $|\cdot|$ as the norm. The only significant detail one must check is that the proof of Theorem 2 uses the regret statement for each sub-algorithm as a black box, and so the nonlinearity of the comparator $\mathcal{F}$ does not change the analysis. $\qquad\square$

**Proof of Theorem 8.** This is a corollary of Theorem 7. That theorem, configured with one sub-algorithm for each class $\mathcal{F}_k$ and with $L_k = L$, $R_k = R_k$, and $\pi_k = 1/k^2$, implies

$$\mathbb{E}\left[\sum_{t=1}^{n} \ell(\hat{y}_t^i, y_t) - \inf_{f \in \mathcal{F}_k} \sum_{t=1}^{n} \ell(f(x_t), y_t)\right] \leq \mathbb{E}[\mathbf{Rad}_n(\mathcal{F}_k)] + O\left(R_k L \sqrt{n \log(R_k L n k)}\right) \quad \forall i \in [N].$$

(20)

The final regret bounded stated follows from the assumed growth rate on $\mathbf{Rad}(\mathcal{F}_k)$. □

**Proof of Theorem 9.** We briefly sketch the construction as follows:

1. For each $\mathcal{H}_k$, construct a sequence of nested subclasses (norm balls) as precisely as in the proof of Theorem 3. There will be $O(n)$ sub-algorithms for each such class.

2. For each sub-algorithm in class $k$, take the prior weight $\pi$ proportional to $1/nk^2$.

Using the analysis from Theorem 3 — namely that for each norm $\|\cdot\|_{\mathcal{H}_k}$ it is sufficient to only consider predictors with norm bounded by $e^n$ —, one can see that the result follows from Theorem 7. □