[Reviews · NeurIPS 2017]

Reviewer 1



Online learning has recently attracted much attention due to its involvement as stochastic gradient descent in deep learning. Oracle inequalities form an important part of online learning theory. This paper proposes an online algorithm, MultiScaleFTPL, in terms of a scale c_i of regret to expert i. Then an oracle inequality for regret bounds is presented. An essential term is \sqrt{n \log (n c_i/\pi_i)}, which means a tight complexity requirement on the hypothesis set and is caused by the linear design of p_t in Algorithm 3. This special design leads to a restriction of applying the main oracle inequality to supervised learning only to Lipschitz losses in Theorems 8 and 9. But the results in the paper are interesting in general.

Reviewer 2



SUMMARY While I am not heavily familiar with the literature on adaptive online learning, this paper seems to be a breakthrough, offering in the form of Theorem 1 a very strong result that can be leveraged to obtain adaptive (in the model complexity sense) online learning bounds in a number of settings. The efficiency, at least in the polytime sense, of the algorithms for the various settings makes these results all the more interesting. I was very surprised by the aside'' on the 1-mixability of logistic loss and the argument for circumventing the lower bound of Hazan, Koren, and Levy in the supervised learning setting. I wish that the authors could give more detail to this observation and the consequences, as the implications are so interesting that I would be (almost) sold on acceptance from this fact alone. I found the results of this paper to be very interesting, technically strong, and important, so I would strongly recommend acceptance. DETAILED REVIEW This paper is very well written and clearly identifies the core technical contribution. I do not really have any criticisms of the paper. I found Theorem 1 to be a very powerful result, as it readily gives rise to a number of corollaries by appropriately using the MultiScaleFTPL algorithm. I checked part of the proof of Theorem 1 (which is to say I did not look at the proof of Lemma 1), and I think the result is correct. The proof of Theorem 2 appears to be correct, and this result may be thought of as the main bound that can be specialized in the different applications. The ease of the analysis after starting from Theorem 2 is impressive. Regarding the various applications: These all seem either interesting or provide useful examples of how to apply Theorem 2. However, for some of the results (like Theorem 3), even if the previous best results involved inefficient algorithms, it would be good to establish the proper context by providing citations to those results. In the Discussion section, where you discuss Losses with curvature'', do you believe the $\log(n)$ factor arising from the $1/n$-cover to be necessary or just an artifact of using a cover? MINOR COMMENTS Line 154: is the norm $\|\cdot\|_i$ meant to be an $\ell_p$ norm for $p = i$? I suspect not. You are abusing notation right? This level of abuse is too high, and I'd perhaps put parenthese around the subscript $i$ to distinguish from an $\ell_p$ norm. Line 193: convexity chosen'' --> convexity was chosen'' Line 194: incorporate generalize'' doesn't make any sense; fix this

Reviewer 3



The paper introduces a framework for deriving model selection oracle inequalities in the adversarial online learning setting. The framework is based on a new meta-algorithm, called MultiScaleFTPL, which is able to combine base algorithms ("experts"), incurring a regret to the i-th algorithm which is proportional to the range of its losses. This adaptation to the loss range leads to numerous applications of the algorithm in a wide range of settings, in many of which oracle bounds were previously unavailable. In particular, the authors consider adaptation to the comparator's norm in uniformly convex Banach spaces (which is a novel results, as previously only Hilbert space norms were considered), an algorithm with a bound which holds for any \ell_p norm of the comparator (quite surprising that such a generality in the bound is actually possible), adaptation to the rank for online PCA, adaptation to the norm of positive definite comparator matrix in the Matrix Multipcative Weight setup, combining algorithms in the online supervised learning setup, including online penalized risk minimization and online multiple kernel learning. Many of these results are impressive. The overall contribution is thus very solid. The central result in the paper is the MultiScaleFTPL algorithm, essentially a "prediction with expert advice" algorithm. The main feature of the algorithm, the reason why it stands out from other previously considered methods, is its adaptation to the loss range: the algorithm guarantees a regret to the expert i (for all i), which scales proportionally to the range of the losses of expert i, rather than to the worst-case range among experts. The algorithms is an unusual version of Follow the Perturbed Leader algorithm, which samples the future and plays with a distribution which minimizes the worst-case (w.r.t. the current gradient) penalized regret. The technical quality of the paper is high. I verified some of the proofs (not all) in the appendix. I found one confusing aspect in the proof of the main theorem (Thm 1), which I was not able to understand: at the top of page 12 (appendix) what does "choosing p_t to match e_i" mean, given that i is the index over which the supremum is taken? (i.e., it sounds as if p_t matches all all e_i simultaneously...). I would be very to glad to get a feedback from the authors on this issue. In summary, I really like the new algorithms and numerous impressive novel results the algorithm leads to. I think the paper is definitely worth accepting to the conference. Minor remarks: - line 54-55: "all losses ... are all bounded" -> "losses ... are bounded" - Eq. (3) -- what is w_t^i on the l.h.s.? (i index not explained) - line 121: "with norm large as" -> "with norm as large as" - Eq. below line 135: sigma's are denoted differently than in Algorithm 3 - line 187: 2-smooth norm -- this term is not explained (at least give a reference to the definition) - appendix A.1, last line of page 11: < p_t,g_t > -> < p_t, g_t' > - appendix A.1 (proof of Thm 1): would be good to say what \epsilon is. - appendix A.1 (proof of Thm 10): domination is defined by saying that w_1 is at least w_2 plus sum of c_1 + c_2, but there is a factor of 2 in front of epsilon sigma_i c_i in the bound. Should I understand it that epsilon is distributed on {-1/2, 1/2}? But then it does not fit to the proof of Thm 1 (top of page 12) - appendix A.2 (proof of Thm 2): \cal{A} introduced without definition (class of base algorithms?) ------- The rebuttal clarified my (minor) issues with the proofs. I keep my high score.